# Near-Infrared Imaging of Steroid Hormone Activities Using Bright BRET Templates

**DOI:** 10.3390/ijms24010677

**Published:** 2022-12-30

**Authors:** Sung-Bae Kim, Ryo Nishihara, Ramasamy Paulmurugan

**Affiliations:** 1Research Institute for Environmental Management Technology, National Institute of Advanced Industrial Science and Technology (AIST), 16-1, Onogawa, Tsukuba 305-8569, Ibaraki, Japan; 2Health and Medical Research Institute, National Institute of Advanced Industrial Science and Technology (AIST), 1-1-1, Higashi, Tsukuba 305-8566, Ibaraki, Japan; 3Japan Science and Technology Agency (JST), PREST, 4-1-8, Honcho, Kawaguchi 332-0012, Saitama, Japan; 4Molecular Imaging Program at Stanford, Bio-X Program, Stanford University School of Medicine, Palo Alto, CA 94304, USA

**Keywords:** bioluminescence, imaging, bioluminescence resonance energy transfer (BRET), coelenterazine, fluorescent protein, blue-to-red shift, near infrared, luciferase, steroid

## Abstract

Bioluminescence (BL) is an excellent optical readout for bioassays and molecular imaging. Herein, we accomplished new near infrared bioluminescence resonance energy transfer (NIR-BRET) templates for monitoring molecular events in cells with higher sensitivity. We first identified the best resonance energy donor for the NIR-BRET templates through the characterization of many coelenterazine (CTZ)–marine luciferase combinations. As a result, we found that NLuc–DBlueC and ALuc47–nCTZ combinations showed luminescence in the blue emission wavelength with excellent BL intensity and stability, for example, the NLuc–DBlueC and ALuc47–nCTZ combinations were 17-fold and 22-fold brighter than their second highest combinations, respectively, and were stably bright in living mammalian cells for at least 10 min. To harness the excellent BL properties to the NIR-BRET systems, NLuc and ALuc47 were genetically fused to fluorescent proteins (FPs), allowing large “blue-to-red” shifts, such as LSSmChe, LSSmKate2, and LSSmNep (where LSS means Large Stokes Shift). The excellent LSSmNep–NLuc combination showed approximately 170 nm large resonance energy shift from blue to red. The established templates were further utilized in the development of new NIR-BRET systems for imaging steroid hormone activities by sandwiching the ligand-binding domain of a nuclear receptor (NR-LBD) between the luciferase and the FP of the template. The NIR-BRET systems showed a specific luminescence signal upon exposure to steroid hormones, such as androgen, estrogen, and cortisol. The present NIR-BRET templates are important additions for utilizing their advantageous imaging of various molecular events with high efficiency and brightness in physiological samples.

## 1. Introduction

Bioluminescence (BL) has been utilized as a versatile optical readout in assay systems for imaging various molecular events in cells in vitro and cells within living animals [1,2]. The advantage of BL over fluorescence (FL) in molecular imaging is mainly because of a low background signal that results to high signal-to-noise (S/N) ratios, long linear dynamic ranges, and the simplicity in the assay setups because of the unnecessity of an excitation light source for imaging [3,4].

On the other hand, BL commonly suffers from a low absolute intensity, the necessity of the specific substrate, and the limited color palette mostly populated in blue and green, which causes severe tissue attenuation in physiological samples, especially in animal models. Because the tissue attenuation of the blue and green BL is a critical issue, redshifted BL is more appropriate for deep-tissue imaging of molecular events [5]. Such BL drawbacks have been addressed in part by: (i) the synthesis of redshifted coelenterazine (CTZ) analogues [6], (ii) the creation of red-emitting luciferases using genetic engineering [7,8,9], (iii) the synthesis of fluorophore-linked CTZ analogues for through bond energy transfer (TBET) [10], and (iv) optical probes based on bioluminescence resonance energy transfer (BRET) [11,12].

To date, BRET-based assay systems have been well documented in the literature. These systems consist of a luciferase as a resonance energy donor and a fluorescent protein (FP) as the energy acceptor. Various BRET studies with diverse luciferase–FP pairs have greatly contributed to the expansion of the color pallets to the near-infrared (NIR) region. We have previously developed a NIR-BRET system that results to a 300 nm blue-to-NIR shift (400–717 nm) by making use of novel blue-emitting CTZ derivatives and a blue-excited iRFP [11]. This NIR-BRET system was achieved by tuning the emission peak of CTZ derivatives to a Soret band of the iRFP.

In designing new BRET assay systems, one faces two critical choices: (i) the selection of a resonance energy donor and an acceptor pair to exert efficient and bright RET and (ii) the selection of a key molecular event of interest to be imaged using the system.

In this study, we imaged the endocrine hormonal activities of nuclear receptors (NRs) as the target molecular events. This is because of the endocrine systems of NRs, regulating a great number of physiological processes, such as metabolism, reproduction, inflammation, and circadian rhythm [13]. The NRs are endogenously regulated by small lipophilic molecules [13]. For example, sex hormone receptors, such as estrogen receptor (ER) and androgen receptor (AR), are stimulated by specific steroid hormones, namely estrogen and androgen, respectively. Similarly, glucocorticoid receptors (GRs) are activated by cortisols dubbed as “stress hormones”. All these steroid hormones share a similar chemical structure with each other. Some man-made chemicals can mimic or disrupt these endocrine systems and can cause diseases including cancer [14].

Several bioassays have been developed in determining the functional activity of NR hormones. Reporter gene assays were well established [15,16], where a hormone-activated NRs trigger the expression of a reporter protein via a transcriptional regulator. However, these reporter gene assays take a long stimulation time until the reporter is accumulated enough to be determined by instruments.

Protein-fragment complementation assay (PCA) is another methodology that was developed to quantitatively determine the NR hormone activity real time [17,18,19]. In this probe design, a reporter is first dissected into two fragments, and thus temporarily lose their reporter properties. The N- and C-terminal fragments are then genetically fused to the ligand-binding domain (LBD) of an NR (NR-LBD) and its interacting partner protein (or peptide), respectively. If a ligand is eligible to activate the interaction between the NR-LBD and the partner protein, the corresponding fragments are approximated, which reconstitute the luciferase activity and produce signal upon exposure to the substrate luciferase. This strategy is advantageous for imaging the temporal dynamics of the ligand-activated molecular events. However, the problem is that PCA systems recover only 0.5–5% of the original activities. Therefore, they are generally weak in mammalian cells [4].

This present study is intended to develop bright NIR-BRET templates, which are applicable for imaging steroid hormone activities of NRs. We first screened the ideal luciferase–luciferin pairs to develop an excellent resonance energy donor. The chosen energy donors were harnessed into a series of new BRET templates by fusing with the energy donor luciferases to FPs. The constructed BRET templates were exploited to create new NIR-BRET probes for the steroid hormone activities by sandwiching the NR-LBD between the luciferase and FP pairs. The utilities of the developed BRET probes were demonstrated with the determination of the hormonal activities of the three endocrine hormones and their antagonists, namely estrogen, androgen, and cortisol. This NIR-BRET imaging system provides an efficient NIR toolbox for imaging diverse cellular events in living cells without severe optical attenuation and autoluminescence.

## 2. Results and Discussion

### 2.1. BL spectra of Various CTZ Derivatives That Were Close to Each Other at the Blue-Green Region

To date, *Renilla* Luciferase (RLuc) has been the extensively studied luciferase among marine luciferases. It has been well documented that RLuc variants exert excellent optical intensities and color variations. Recently, another smaller marine luciferase, the NanoLuc, was developed by the genetic engineering of an *Oplophorus* luciferase (OLuc) that was derived from a deep-sea shrimp and is currently and actively being used in various imaging studies [20]. We developed a series of artificial luciferases (ALucs) by extracting the frequently occurring amino acids, sequencing them from the database of copepod luciferases, and completing the full sequences [18,19,20,21]. As the corresponding studies are proceeding, it is critically important to determine how a marine luciferase is unique and distinctive in terms of its optical properties compared to other marine luciferases.

We first investigated the BL spectra of native coelenterazine (nCTZ) and its 11 derivatives with selected marine luciferases, namely NLuc, ALuc16 and ALuc47, (Figure 1 and Appendix A). The luciferases, NLuc, ALuc16 and ALuc47, were selected in this study because they are known to be the smallest and brightest marine luciferases according to previous reports [18,20,21].

The results show that the BL spectra populated the blue and green region, and the emission spectra were mostly very close to each other. We did not observe any notable spectral separation from each other. The nCTZ was commonly bright among all the tested marine luciferases, whereas several CTZ analogues were selectively bright with some specific marine luciferases. For example, the DeepBlueC (DBlueC) was exclusively specific to the NLuc, whereas the following CTZ derivatives were specifically bright with ALucs, namely BBlue1.2, BBlue2.1, BBlue2.2, 6-ome-CTZ (Ome), and 6etCTZ. In this experiment, we intentionally chose DBlueC as a substrate of NLuc instead of furimazine (FMZ). This is for the intensity balance among reporters, such as in the case of the NLuc–FMZ combination; it is too bright to be used in our multi-reporter systems and can bury other signals.

The absolute intensities and wavelengths of the spectral peaks (λ_max_) were summarized in Appendix A. The most blue-shifted λ_max_ was observed at 447 nm with the BBlue2.4–NLuc combination, whereas the most redshifted signal was observed at 498 nm with BBlue1.2–ALuc16 and Ome–ALuc16 combinations. The highest and second highest peak intensities were determined with nCTZ–ALuc47 and DBlueC–NLuc combinations, respectively. Because the spectra mostly populated the blue and green regions, spectral portions longer than 600 nm occupy merely less than 13% of the total spectral area.

These blue- and green-populated BL signals can be severely attenuated by the hemoglobin and other ingredients in physiological samples [22]. Moreover, such short-wavelength and band-broaden BL signals are difficult to be unmixed when they are used in multi-reporter systems. This feature is a great demerit, considering that multi-reporter systems generally reduce measurement time and expense and provide collective information with high fidelity [23].

The full width half maximum (FWHM) values of the representative substrate–luciferin combinations, namely nCTZ–NLuc and nCTZ–Aluc16, were found to be 90 and 102 nm, respectively. It was previously studied that during the BL reaction, nCTZ can form four possible intermediates in different protonation states (e.g., neutral species, amide anion, phenolate anion, and pyrazine anion species), which determine the emission colors [24,25,26]. The anion intermediates contribute to the longer wavelength peaks (480–565 nm), whereas the neutral one works for the shorter wavelength peak (386–423 nm). In considering this view, the small FWHM values and peak positions in green implicate that the major BL spectra of nCTZ with NLuc, ALuc16, and ALuc47 in Figure 1 were mostly contributed by a singular intermediate species at the middle energy level (i.e., phenolate anion, 480–490 nm).

### 2.2. Some Luciferins Are Exclusively Bright with Mammalian Cells Stably Expressing CBLuc Red, NLuc or ALuc47

We characterized the reactivity of 12 different substrates with live COS-7 cells stably expressing CBLuc Red, NLuc, ALuc16, and ALuc47 (Figure 2 and Appendix A).

The results show that all the 12 types of CTZ derivatives failed to generate BL intensities with the CBLuc Red, whereas DBlueC and nCTZ were exclusively bright with NLuc and ALuc47, respectively (Figure 2A). The time course showed that the DBlueC–NLuc and nCTZ–ALuc47 pairs luminesce with the very stable BL intensities during the initial 10 min of the assay time in live COS-7 cells, namely the BL intensities of DBlueC–NLuc and nCTZ–ALuc47 pairs in the live cells maintained up to 94 and 46% of the maximal intensities, respectively, 10 min after the substrate injection. The peak BL intensities of DBlueC–NLuc and nCTZ–ALuc47 pairs showed at least 17- and 22-fold stronger signals than the peaks of the other pairs, respectively.

The left and right panels of Appendix A show the comparison in the BL intensity profiles by the time of the live COS-7 cells and the lysate, respectively. The profiles may be summarized as follows: (i) the CBLuc Red does not luminesce with all the tested CTZ analogues, (ii) the NLuc in live cells is exceptionally bright with DBlueC but largely weakens the specificity in lysates, (iii) the ALuc16 shows relatively weak BL intensity profile in live cells but boosts the absolute BL intensities in lysates, (iv) the ALuc47 is exceptionally bright only with nCTZ in live cells and maintains the specificity even in lysates.

Except in the case of the NLuc–DBlueC pair, all the other pairs showed boosted BL intensities in lysates, as compared to live cells. This result may be interpreted with the plasma membrane permeability of each CTZ analogues in live cells [27]. In living mammalian cells, ALuc16 and ALuc47 are sequestered into the endoplasmic reticulum (ER) because they are tagged with a “KDEL” sequence to suppress their intrinsic secretive nature in this experiment. This means that the CTZ analogues must gain access to ALuc16 and ALuc47 in the ER after crossing through the plasma membrane barriers. Even in the case of NLuc, they are mostly located in the cytosol, and thus the CTZ analogues need to cross only the plasma membrane to reach NLuc in live cells. Hence, any hydrophilic functional groups in the CTZ analogues may hamper the membrane’s permeability to reach marine luciferases.

The accessibility of the substrates to the marine luciferases is greatly improved without the presence of plasma membrane barrier in cell lysates. Hydrophilic functional groups such as the hydroxy groups in the substrates can hamper the plasma membrane’s permeability. We have previously shown that the lipid-water partition coefficients (*p*) of selected substrates [11] act as a direct reference to assume the plasma membrane permeability of the substrates. The only example that is against this view is the case of the DBlueC–NLuc pair, which is exclusively bright in live cells but greatly suppressed in the lysate (Appendix A). The reason for the reduction in the NLuc activity upon lysing the cells in this lysis buffer (cat. E2820, Promega) is due to the acidic pH level of the applied buffer.

### 2.3. Microslides Highlight Luciferase Specificity of DBlucC and nCTZ

The outstanding luciferin specificity of marine luciferases was further highlighted using six-channel microslides by growing COS-7 cells, which express the CBLuc Red, NLuc, or ALuc47 (Figure 2B). It confirms that the DBlueC exclusively illuminates the channels of the cells containing the NLuc, whereas the nCTZ selectively enhances the BL intensities of the cell channels containing the ALuc47. On the other hand, both the DBlueC and nCTZ commonly failed to illuminate the cell channels expressing the CBLuc Red.

The comparison of the fold intensities shows that the nCTZ illuminates 79-fold stronger BL with the ALuc47 than with the CBLuc Red. The DBlueC is also 26-fold brighter with the NLuc than with the CBLuc Red. On the other hand, we found that the D-luciferin luminesces with the CBLuc Red, which is 5-fold stronger than with the ALuc47. These results convince us that we can create a multiplex imaging system using such luciferase–luciferin combinations.

### 2.4. The pH Determines the Specificity of Bioluminescence of CTZ Analogues by Reducing the Autoluminescence

We further examined the pH-dependent autoluminescence of selected CTZ analogues using a universal buffer (Appendix A). The results show that the autoluminescence intensities have a tendency to be suppressed in the low pH range (acidic condition) and which gradually increases in the high pH range (basic condition). The BBlue2.1, BBlue1.6, and Ome are especially unstable in the basic buffer conditions, for example, the autoluminescence intensity of the BBlue1.6 at pH 10 was five-fold higher than at the same pH of 7.

The BL intensities in the presence of a marine luciferase, namely ALuc16 or ALuc47, were also greatly influenced by the pH levels of the universal buffer (Appendix A). In the case of the ALuc16, the BL intensities were at the weakest at pH 5, which can be gradually increased by raising the pH levels and achieved the highest signal at pH 8 or 9 depending on the substrates. The BL intensity of the ALuc47 also showed a similar pH dependency, and the signal peaked at pH 8 with nCTZ, for example, the BL intensity of the ALuc47–nCTZ pair in pH 8 was 21-fold stronger than that in pH 5. The only difference of the BL intensity profile of the ALuc47 compared with the ALuc16 was the exclusive specificity to nCTZ at any pH levels.

### 2.5. New BRET Systems Luminesce with Characteristic NIR-BL

The above-mentioned studies on luciferin-luciferase reactions inspired us to fabricate new BRET-imaging systems emitting characteristic NIR-BL with luciferin specificity (Figure 3). Because the ALuc47 and NLuc exclusively reacted with nCTZ and DBlueC, respectively, we created six different BRET templates by genetically fusing the luciferases with fluorescent proteins (FPs). Among the many FP candidates, we chose the LSSmChe, LSSmKate2, and mNeptune with the C162D mutation (named LSSmNep hereafter) because their excitation spectra are well overlapped with the emission spectra of the ALuc47 and NLuc, the FPs allow large Stokes shifts (LSSs) between the excitation and emission spectra, which minimize the background intensities, and the FPs have better quantum yields than the other candidates.

We first examined whether the BRET templates can generate expected BRET peaks in the red and NIR regions (Figure 3B). The results with 640 and 700 nm bandpass filters show that the LSSmNep_NLuc emits the brightest BL and followed by the LSSmKate2_NLuc and LSSmNep_A47. This feature was commonly observed in live cells, as well as in the lysates.

The BRET signal was confirmed by the unique spectra (Figure 3C). Because the LSSmNep_NLuc was the brightest, the BRET spectra were further determined with the DBlueC dissolved in universal buffers or assay buffers (Promega). The spectra show the characteristic peaks at 470, 477, and 640 nm. It is interpreted that the peaks at 470 and 477 nm denote the maximal intensities (λ_max_) of NLuc as the resonance energy donor, whereas the peak at 640 nm confirms the emission of the LSSmNep as the energy acceptor. The spectral portion longer than 600 nm was ca. 30% of the total spectral area.

### 2.6. New BRET Imaging Probes Quantitatively Image the Activities of Steroid Hormones

Because LSSmNep_NLuc and LSSmNep_ALuc47 showed the brightest NIR-BRET profiles, we further investigated whether the NIR-BRET templates work as an efficient tool for imaging steroid hormones (Figure 4 and Figure 5). In specific, the AR HLBD, GR HLBD, ER HLBD, and ER LBD were genetically sandwiched between the FP and luciferase of the templates with various GS linkers and made 15 types of new BRET probes.

We investigated the BRET ratios (i.e., intensity at 640 nm/intensity at 500 nm) of the new BRET probes before and after stimulations using steroid hormones. The results were summarized in Figure 5. Among the AR probes, the LSSmNep-AR-NLuc showed an ~19% increase in BL intensity after an androgen stimulation in live cells. Among the GR probes, the LSSmNep-GR5GS-NLuc and LSSmNep-GR9GS-NLuc exerted a 32% and 27% increase in BL intensities, respectively, after a cortisol stimulation in live cells. This feature was also observed in the lysates. Among the ER probes, the LSSmNep-SH2-ER-NLuc showed a 16% enhancement in BL intensity in live cells after an estrogen antagonist (OHT) stimulation. This feature was similarly observed in the lysates.

The overall results may be summarized as follows: (i) the LSSmNep_NLuc system makes a universal BRET template for imaging conformation changes of proteins of interest or protein–protein interactions (PPIs) in cells, and (ii) in considering that the only difference between the probes is the sandwiched NR-LBD, the enhanced NIR-BL can be explained as the inserted NR-LBD modulates the BRET efficiency by varying the relative distance and orientation between the LSSmNep and NLuc.

We further investigated whether the selected BRET probes improve their specificity to distinguish between agonists and antagonists with NIR-BLI (Figure 6). The results show that the LSSmNep-SH2-ER-NLuc in live cells exclusively enhanced the NIR-BRET ratios up to 3.1-fold with both of its agonist, E2, and antagonist, OHT. However, the probe did not enhance any considerable NIR-BRET ratios with progesterone, although progesterone chemically resembles with E2. Furthermore, the LSSmNep-AR-NLuc in live cells exclusively elevated its NIR-BRET ratio up to 1.7-fold in the presence of its known agonist, DHT. The LSSmNep-GR5GS-NLuc in live cells also exerted 1.3-fold enhanced NIR-BRET intensities only with its agonist, cortisol. This agonist specificity confirms that it did not show any elevated NIR-BRET ratio with other steroids.

A similar ligand specificity of the probes was observed even in the lysates (Figure 6B). The LSSmNep-SH2-ER-NLuc in lysate showed a 4.0-fold and 3.1-fold elevation in the NIR-BRET ratios with OHT and E2, respectively. The LSSmNep-AR-NLuc in lysate also shows 2.2-fold enhanced NIR-BRET ratios only with the agonist DHT. The same probe slightly enhanced the ratios with flutamide and E2 but not as much as with DHT. It is considered that the flutamide and E2 worked as a partial agonist for the LSSmNep-AR-NLuc.

The overall result is interpreted as follows: (i) the sandwiched NR-LBD in the BRET templates exclusively recognizes its specific ligand from the other steroids, (ii) the unique ligand specificity of the selected probes in live cells was also observed even in the lysates, and (iii) the working mechanism of the LSSmNep-SH2-ER-NLuc is supposed as follows: the OHT-activated ER-LBD is phosphorylated at a specific site, which is recognized by the adjacent SH2 domain. This ER LBD-SH2 interaction exerts a structural fold-up, and thus approximates the adjacent proteins, namely LSSmNep and NLuc. This approximation has resulted in improved BRET efficiency as illustrated in Figure 6C, inset ‘*a*’. The E2 is known to trigger both the phosphorylation and transformation of the α-helix 12 of the ER-LBD. This working mechanism regarding the BRET efficiency-distance correlation was well documented in the precedent studies [17,28,29].

### 2.7. Dose–Response Curves of the NIR BRET Probes for Steroids Confirm their Ligand Specificity and Detection Limit

We further determined the dose–response curves of LSSmNep-SH2-ER-NLuc and LSSmNep-AR-NLuc, because the selected BRET probes showed relatively high NIR-BRET ratios in the above studies (Figure 6C).

The results show that LSSmNep-SH2-ER-NLuc exclusively elevated the NIR BL intensities in response to OHT and E2, but not DHT, when a 700 nm bandpass filter was applied for the measurement. The detection limits to OHT and E2 were found to be 1.6 × 10^−8^ M and 1.9 × 10^−7^ M, respectively. On the other hand, LSSmNep-AR-NLuc was also highly specific only to DHT and the detection limit was measured to be 4.4 × 10^−7^ M.

## 3. Methods and Materials

### 3.1. Design and Synthesis of CTZ Analogues

For the present study, a series of CTZ analogues were synthesized with modification of the C-6 position. The hydroxy group at the C-6 position of native CTZ is replaced with various functional groups like ether, alkyl, or ethyl groups.

The specific synthetic process of the CTZ analogues is described in Experimental Procedure S1 in detail. The chemical structures are highlighted in Appendix A.

### 3.2. Measurement of BL Spectra of CTZ Derivatives According to Marine Luciferases

The following mammalian expression vectors were obtained from our precedent studies: pcDNA3.1(+) vectors encoding NanoLuc, Artificial Luciferase 16 (ALuc16), or Artificial Luciferase 47 (ALuc47).

COS-7 cells derived from African green monkey kidney fibroblast were cultured in 6-well microplates and grown until the population reached 80% confluency. The cells were transiently transfected with 2.5 mg/mL pcDNA3.1(+) vector per well encoding NanoLuc, Artificial Luciferase 16 (ALuc16), or Artificial Luciferase 47 (ALuc47) using a lipofection reagent, TransIT-LT1 (Mirus), according to the manufacturer’s instruction. The cells were further incubated for two days in a 5% (*v*/*v*) CO_2_ incubator. The cells on each well were lysed using a passive lysis buffer (Promega), and 40 μL of the lysates were aliquoted into 200-μL PCR tubes. On the other hand, native CTZ and its derivatives were first dissolved in methanol and then diluted to 0.1 mM using 1× phosphate-buffered saline (PBS), named as “substrate solution”.

We injected 40 μL of each substrate solution into the PCR tube and the tube was immediately mounted in the sample stage of a high-precision spectrophotometer (AB-1850, ATTO). The corresponding BL spectra were determined in the high sensitivity mode (Figure 1B and Appendix A).

### 3.3. Determination of BL Intensities of the CTZ Derivatives According to Marine Luciferases

The absolute BL intensities (BLI) of all CTZ derivatives were simultaneously determined against selected marine luciferases, i.e., Click Beetle luciferase red (CBLuc red), NanoLuc, and ALuc47 (Figure 2A).

COS-7 cells were cultured in 6-well plates and transiently transfected with pcDNA3.1(+) vector encoding CBLuc red, NanoLuc, or ALuc47 using a lipofection reagent, TransIT-LT1 (Mirus). The cells were further incubated for 1 day in a 5% (*v*/*v*) CO_2_ incubator and subcultured into 96-well black wall clear bottom microplates (Nunc) or 6-channel microslides (ibidi, Gräfelfing, Germany). The cells were further incubated overnight in the CO_2_ incubator for stabilization.

The culture media in the wells of the microplates were aspirated completely, and the wells were simultaneously injected by 40 μL of the substrate solutions containing one of the following substrates using a multichannel micropipette: i.e., CTZ, BBlue1.2, BBlue1.6, BBlue2.1, BBlue2.2, BBlue2.4, BBlue3.1, BBlue3.2, 6-ome-CTZ, 6-pi-Ph-CTZ, CTZi, and DBlueC. The corresponding BL images from the live cells were determined using an IVIS optical imaging system. The BL signal was quantitatively analyzed using a specific software, Living Image software ver. 7.4, by drawing region of interest (ROI) over the images.

Likewise, the culture media in the channels of the microslides were completely removed, and the channels were simultaneously filled with 60 μL of the substrate solution containing CTZ or DBlueC. The consequent BL images from the live cells in the microslides were immediately determined using the IVIS optical imaging system, and were quantitatively analyzed using Living Image software ver. 7.4.

The pH-dependency of the BL intensities of the substrates was also determined with Universal buffers (Appendix A). We prepared the universal buffer with the composition of 0.05 M citric acid, 0.2 M boric acid, and 0.1 M in Na_3_PO_4_ as the final concentrations in distilled water and adjusted it to the required pH using 0.1 N NaOH solution. COS-7 cells were grown in 6-well plates and transiently transfected with pcDNA3.1(+) vector encoding ALuc16 or ALuc47. The cells were incubated for 2 days in a CO_2_ incubator, and sub-cultured into 96-well black wall microplates and further incubated overnight. Separately, a series of the substrate solutions were prepared to be 100 μM through dissolving nCTZ into Universal buffer with the pH ranging from 5 to 10. The culture media in the wells were completely aspirated, and the wells were simultaneously injected with 40 μL of the substrate solutions with different pHs. The corresponding BL intensities were immediately determined using the IVIS optical imaging system and quantitatively analyzed using Living Image ver. 7.4.

### 3.4. Determination of Autoluminescence of Selected Substrates According to pH

The autoluminescence properties of selected substrates were examined in Universal buffers of various pHs ranging from pH 5 to 10 (Appendix A). The stock solutions of the substrates were diluted into the Universal buffers to a final concentration of 100 μM. Forty microliters of the diluted solutions were aliquoted into the wells of a 96-well black wall microplate. The corresponding BL images were determined using the IVIS optical imaging system. The signals were quantified using Living Image software ver. 7.4 by drawing region of interest (ROI) over the image.

### 3.5. Construction of Mammalian Expression Plasmids for Fundamental NIR-BRET Imaging

Six different plasmids encoding near infrared (NIR)-BRET probes were constructed for the mammalian expression (Figure 3A).

The cDNA templates encoding fluorescent proteins (FPs), i.e., LSSmCherry, LSSmKate2 (S159T), LSSmNep (i.e., mNeptune with C162D mutation), and mCherry, were custom-synthesized by Eurofins Genetics (Tokyo, Japan) according to the precedent sequence information [30,31].

First, a series of cDNA segments encoding three different fluorescent proteins (FP) were generated by PCR using corresponding primers to introduce unique restriction sites, *Hind*III/*BamH*I. Similarly, cDNA segments encoding ALuc47 (19–193 AA) and NanoLuc were synthesized by PCR using corresponding primers to introduce *BamH*I (*Kpn*I)/*Xho*I at the 5′ and 3′ ends, respectively. The cDNA segments were double digested by the corresponding restriction enzymes (NEB, Ipswich, MA, USA), ligated using a ligation kit (Takara Bio), and subcloned into a respective enzyme-digested pcDNA3.1(+) vector (Invitrogen, Carlsbad, CA, USA) as illustrated in Figure 3A. After expression, the probes were named LSSmChe_A47, LSSmKate2_A47, LSSmNep_A47, mChe_NLuc, LSSmKate2_NLuc, and LSSmNep_NLuc for highlighting the components.

We used N-terminal secretion peptide (SP: 1–18 AA)–eliminated ALuc47 (i.e., 19–193 AA) in this study for excluding spontaneous secretion of the expressed fusion proteins from cells.

### 3.6. Characterization of NIR-BRET Imaging Probes in Mammalian Cells

COS-7 cells were first cultured in 6-well plates until the cells reached a 70% confluence and separately transfected with one of the plasmids encoding LSSmChe_A47, LSSmKate2_A47, LSSmNep_A47, mChe_NLuc, LSSmKate2_NLuc, or LSSmNep_NLuc. The cells were incubated further for 2 days in 5% (*v*/*v*) CO_2_ incubator and subcultured into 96-well black wall microplates. The cells in the microplates were further incubated overnight before using them for imaging.

The wells of each probe were then divided into two sections: one section is for live cell imaging, and the other is for post-lysis imaging. The culture media in the wells were completely aspirated. The wells for live cell imaging remained as is, whereas each well for lysis imaging was lysed with 40 μL of a lysis buffer (Promega) for 15 min.

Both sections were simultaneously injected with 40 μL of the substrate solution containing nCTZ or DBlueC using a multichannel micropipette, and the corresponding BL images were acquired using the IVIS optical imaging system as shown in Figure 3B. The BL images were quantified using the Living Image software ver. 7.4.

The corresponding BRET spectra were determined with the cells containing LSSmNep_NLuc (Figure 3C). COS-7 cells grown in a 6-well microplate were transiently transfected with pcDNA3.1(+) vector encoding LSSmNep_NLuc and incubated for 2 days in a 5% CO_2_ incubator. The cells were then lysed with a lysis buffer (Promega) for 15 min. We aliquoted 40 μL of the lysate to a 200 μL PCR tube. Separately, we prepared substrate solutions to a concentration of 100 μM by dissolving the DBlueC stock solution into an assay buffer (Promega), Universal buffer of pH 8, or Universal buffer of pH 9. After injection of each substrate solution into the PCR tubes, the BL spectra were immediately collected using a precision spectrophotometer (AB-1850 spectrophotometer, ATTO, Tokyo, Japan). BL spectra were measured at 0.6 nm increments from 391 to 790 nm.

### 3.7. Construction of Mammalian Expression Plasmids for Steroid Hormone-Activatable NIR-BRET Imaging

Inspired by the excellent BRET efficiency of the probes, LSSmNep_A47, LSSmKate2_NLuc, and LSSmNep_NLuc (Figure 3B,C), the corresponding cDNA constructs were further modified to create 15 different single-chain probes for imaging steroid hormones interaction with the receptor as shown in Figure 4.

We first custom-synthesized various cDNA segments encoding the following domains by order to Eurofins Genetics (Tokyo, Japan), where the 5′ and 3′ ends were designed to commonly have *BamH*I and *Kpn*I sites, respectively: i.e., the hinge and ligand binding domain of human androgen receptor (AR HLBD; 628–923 aa), 5- and 9-GS linker-fused AR HLBDs, the hinge and ligand binding domain of human estrogen receptor (ER HLBD; 281–549 aa), 4- and 9-GS linker-fused ER HLBDs, the hinge and ligand binding domain of human glucocorticoid receptor (GR HLBD; 486–777 aa), and 5- and 9-GS linker-fused GR HLBDs. On the other hand, the cDNA segments encoding SH2-domain-fused ER LBD and LXXLL-motif-linked GR LBD were obtained from our precedent studies [17] [32].

The created cDNA segments were double digested by the corresponding restriction enzymes (NEB, Ipswich, MA, USA), ligated using a ligation kit (Takara Bio) into respective enzyme-digested pcDNA3.1(+) vectors (Invitrogen, Carlsbad, CA, USA) encoding LSSmNep-fused ALuc47, LSSmKate2-fused NanoLuc, or LSSmNep-fused NanoLuc with various Glycine-Serine (GS) linkers as illustrated in Figure 4. After expression, the probes were named as follows: ① LSSmNep-ARHLBD-NLuc, ➁ LSSmKate2-ARHLBD-0GS-NLuc, ➂ LSSmKate2-ARHLBD-5GS-NLuc, ➃ LSSmKate2-ARHLBD-9GS-NLuc, ➄ LSSmNep-ARHLBD-A47, ➅ LSSmKate2-GRHLBD-NLuc, ⑦ LSSmNep-GRLBD-LXX-NLuc, ⑧ LSSmNep-GRLBD-5GS-NLuc, ⑨ LSSmNep-GRLBD-9GS-NLuc, ⑩ LSSmNep-GRHLBD-A47, ⑪ LSSmKate2-ERHLBD-NLuc, ⑫ LSSmNep-ERHLBD-4GS-NLuc, ⑬ LSSmNep-SH2-ERLBD-NLuc, ⑭ LSSmNep-ERHLBD-4GS-A47, and ⑮ LSSmNep-ERHLBD-9GS-A47 for highlighting the components. The probes ①–➄ are grouped into AR probes because they are designed for androgen imaging, whereas the probes ➅–⑩ are categorized into GR probes as they are created for cortisol imaging. On the other hand, the probes ⑪–⑮ are named ER probes since they are designed for estrogen imaging.

The fidelity of all the cDNA constructs was confirmed with a genetic sequence analyzer (Applied Biosystems, Waltham, MA, USA).

### 3.8. Characterization of the Steroid Hormone Probes with Respect to the BRET Efficiency

COS-7 cells grown in 6-well plates were separately transfected with the pcDNA3.1(+) vectors encoding the single chain steroid hormone probes ①–⑮ using a TransIT-LT1 transfection reagent (Mirus) (Figure 5). The cells were incubated for 24 h in a 5% (*v*/*v*) CO_2_ incubator and subcultured into 96-well black wall microplates. The cells transfected with AR probes were stimulated for 1 h with 10^−6^ M DHT or the vehicle (0.1% (*v*/*v*) DMSO), whereas the cells transfected with GR probes were stimulated for 1 h with 10^−6^ M cortisol or its vehicle (0.1% (*v*/*v*) DMSO). On the other hand, the ER probe-transfected cells were stimulated for 1 h with 10^−6^ M 17β-estradiol (E2).

The wells of each probe were then divided into two sections: one section is for live cell imaging, and the other section is for post-lysis imaging. The culture media in the wells were completely removed. The wells for live cell imaging remained as is, whereas each well for imaging cell lysates was further lysed using 40 μL of a passive lysis buffer (Promega) for 15 min.

Both sections were simultaneously injected with 40 μL of the substrate solution using a multichannel micropipette and the corresponding BL images were determined with the IVIS optical imaging system. The BL images were quantified with the specific software, Living Image 7.4.

### 3.9. NIR-BRET Imaging with Selected Steroid Hormone Probes

COS-7 cells were cultured in 10 cm cell culture dishes and transfected with pcDNA3.1(+) vector encoding one of the steroid hormone probes ①, ⑧, ⑬, ⑭, ⑮, where these probes were chosen from the results of Figure 5. The cells stably expressing the probes were established by consecutive subculturing and screening them with the culture media containing G418.

Three weeks later after the screening, the cells expressing the steroid hormone probes were subcultured into 96-well black wall microplates. The culture media of the cells were completely removed and stimulated with the culture media containing 10^−6^ M of 4-hydroxytamoxifen (OHT), E2, progesterone, cortisol, dihydrotestosterone (DHT), flutamide or the vehicle (0.1% (*v*/*v*) DMSO) for 1 h. After stimulation, the culture media were completely removed and used for imaging. The half of the wells stayed as is, whereas the other half of the wells were used for lysing with the lysis buffer (Promega) for 15 min.

The wells were then simultaneously injected with 40 μL of an assay buffer (Promega) containing nCTZ or DBlueC. The corresponding BL images were acquired using the IVIS optical imaging system equipped with a 700 nm band pass filter. The results were quantitatively analyzed using the Living Image software ver. 7.4.

The corresponding dose–response curves were determined using the cells stably expressing the probes ① or ⑬ (Figure 6C). The cells stably expressing the probe ① or ⑬ were established as described above and subcultured into 96-well black wall microplates. After overnight incubation, the cells in the wells were stimulated with the culture media containing varying concentrations of E2, OHT, or DHT ranging from 10^−10^ to 10^−5^ M, besides the vehicle (0.1% (*v*/*v*) DMSO) for 1 h. The culture media were then completely removed, and the cells were injected with 40 μL of the assay buffer containing DBlueC. The optical images of the microplates were immediately acquired using the IVIS optical imaging system equipped with a 700 nm band pass filter, and the results were quantitatively analyzed using Living Image software ver. 7.4.

## 4. Conclusions

In summary, we created excellent NIR BRET templates that can be easily adopted for imaging various molecular events in cells and animal models. We first characterized the optical properties of many luciferin–luciferase combinations and found that NLuc–DBlueC and ALuc47–nCTZ combinations luminesce in green with excellent BL intensity, specificity, and stability. The excellent BL properties were harnessed to design new NIR BRET systems, where NLuc and ALuc47 were genetically fused to FPs allowing large *blue-to-red* shifts such as LSSmChe, LSSmKate2, and LSSmNep. The new NIR BRET systems were further applied for imaging steroid hormones activities through sandwiching the NR LBD between the luciferase and the FP of the template as a single chain fusion protein. The present NIR BRET templates are important additions to the existing list of BRET probes for advantageous imaging of molecular events with high efficiency and brightness. The NIR BRET templates open broad future applications to quantitative imaging of various molecular events with excellent tissue permeability in physiological samples and animal models.

## Figures and Tables

**Figure 1 ijms-24-00677-f001:**
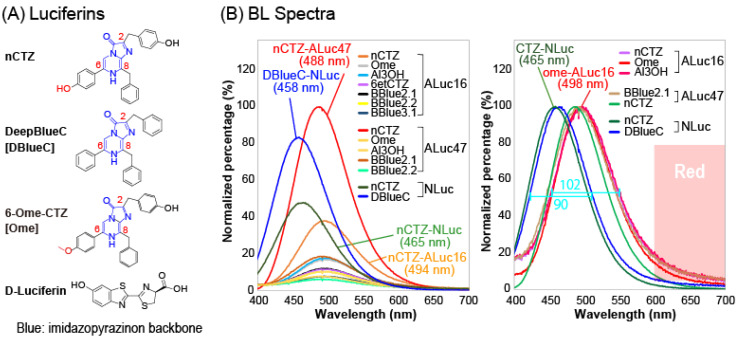
(**A**) Chemical structures of conventional or newly synthesized CTZ derivatives: nCTZ, native coelenterazine; DBlueC, DeepBlueC; Ome, 6-ome-CTZ. They commonly share an imidazopyrazinone backbone that is highlighted in blue. (**B**) Relative and normalized BL spectra of selected CTZ derivatives with marine luciferases, ALuc16, ALuc47, and NanoLuc (NLuc). The representative full width at half maximum (FWHM) values were annotated with cyan color.

**Figure 2 ijms-24-00677-f002:**
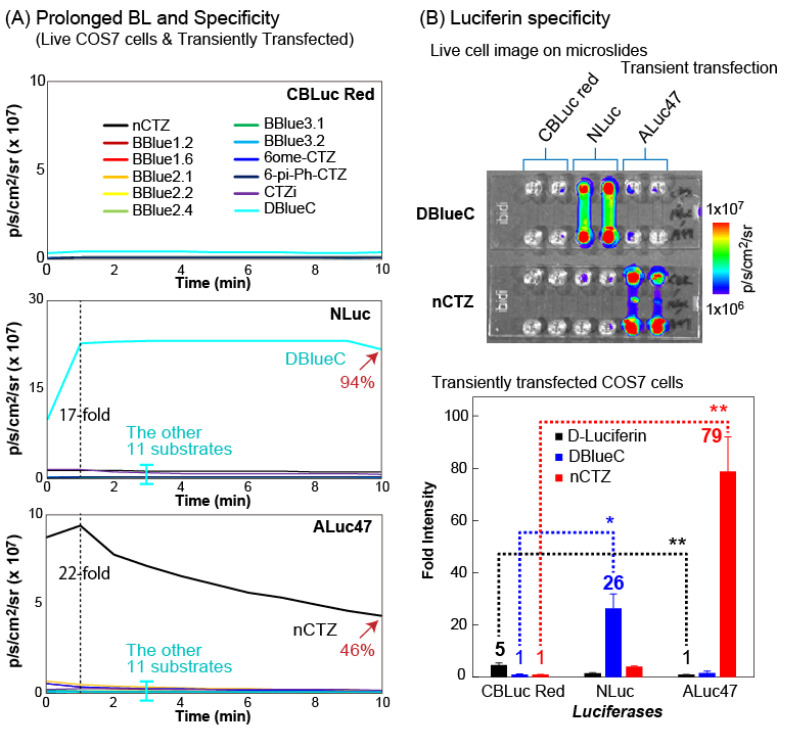
(**A**) The time course of the BL intensities of selected marine luciferases reacted with conventional and newly synthesized CTZ derivatives. (**B**) The BL images of live COS-7 cells stably expressing click beetle luciferase red (CBLuc red), NLuc, or ALuc47 on a 6-channel microslide in the presence of DBlueC or CTZ (upper panel). The corresponding fold intensities of the COS-7 cells highlighting their substrate specificity (lower panel). The p-values (student’s *t*-test) are ** < 0.01 and * < 0.1, respectively.

**Figure 3 ijms-24-00677-f003:**
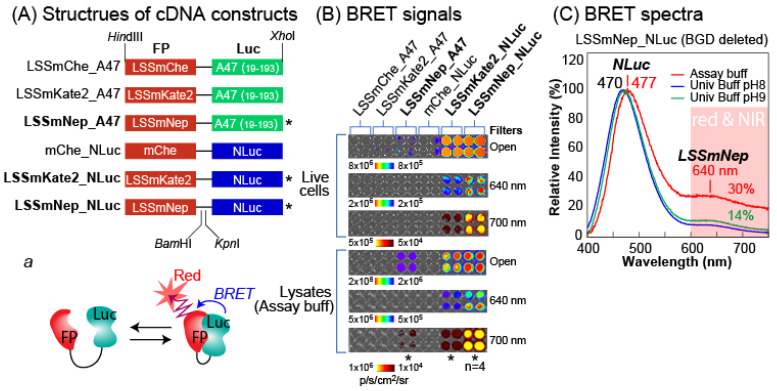
(**A**) Schematic diagram of cDNA constructs encoding various combinations of fluorescent protein (FP) and luciferase, designed for BRET imaging. Inset ‘*a*’ shows the typical working mechanism of the BRET probes after expression. (**B**) The BL images of various BRET probes in the presence of the specific substrate, CTZ for ALuc47-bearing probes (left side), or DBlueC for NLuc-bearing probes (right side). The images were captured with open, 640-nm, or 700 nm band-pass filter (*n* = 4). The asterisks (*) highlight brighter probes. (**C**) The BL spectra of LSSmNep_NLuc in various buffer conditions. The red shadow highlights the red and NIR region of the spectra.

**Figure 4 ijms-24-00677-f004:**
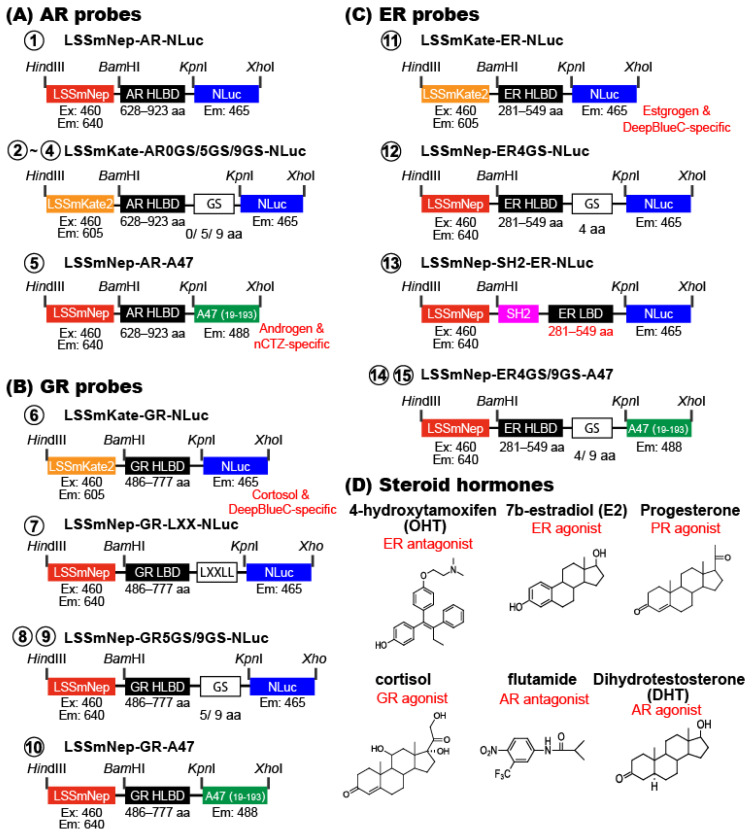
cDNA constructs encoding various BRET probes for imaging steroid hormones. (**A**) A schematic diagram of BRET probes sandwiching AR HLBD. (**B**) A schematic diagram of BRET probes sandwiching GR HLBD. (**C**) A schematic diagram of BRET probes sandwiching ER HLBD. (**D**) The chemical structures of steroid hormones and their antagonists.

**Figure 5 ijms-24-00677-f005:**
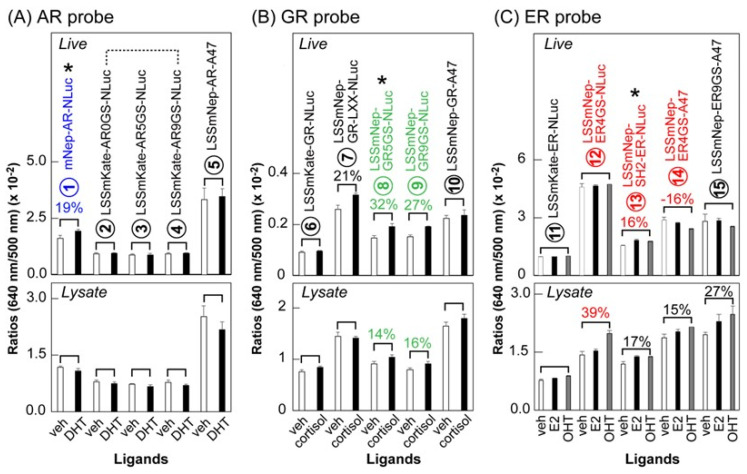
Initial screening of potential BRET probes for imaging steroid hormones. The upper panel shows the optical responses of each BRET probe in live COS-7 cells, and the lower panel shows the same in cell lysates. (**A**) Variance in the optical intensities of BRET probes sandwiching AR HLBD, stimulated by vehicle or DHT. (**B**) Variance in the optical intensities of BRET probes sandwiching GR HLBD, which were stimulated by vehicle or cortisol. (**C**) Variance in the optical intensities of BRET probes sandwiching ER HLBD, which were stimulated by vehicle, E2, or OHT. Abbreviations: AR HLBD, The hinge and ligand binding domain of human androgen recepter; ER HLBD, The hinge and ligand binding domain of human estrogen recepter; GR HLBD, The hinge and ligand binding domain of human glucocorticoid recepter; 0-9GS, Glycine and serine linker in the size of 1–9; LXX, An LXXLL motif (where L is leucine and X is any amino acid); veh, vehicle; DHT, 5α-dihydrotestosterone; E2, 17β-estradiol; and OHT, 4-Hydroxytamoxifen. The asterisks (*) highlight the best S/B ratio probes in the groups.

**Figure 6 ijms-24-00677-f006:**
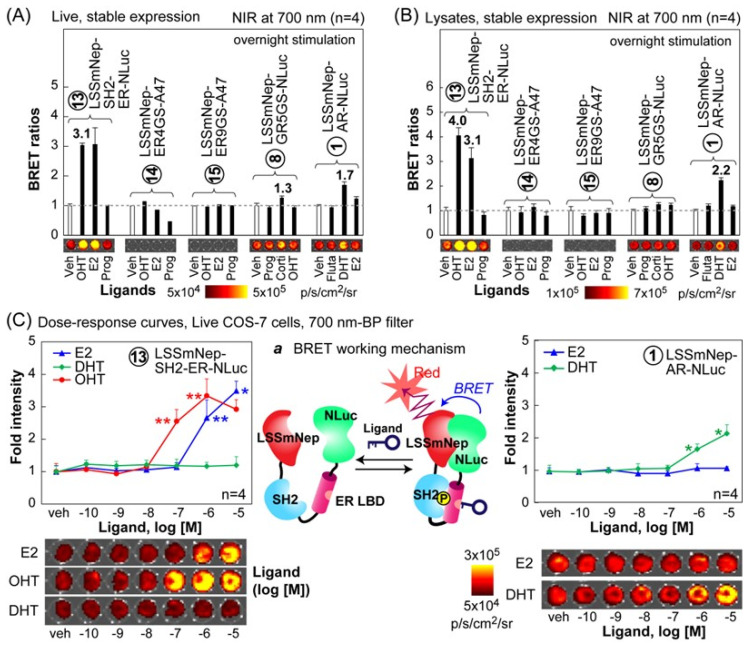
Ligand specificity of selected NIR BL imaging probes stably expressed in mammalian cells. (**A**) NIR BL intensities of the selected imaging probes in live cells after stimulation using various steroid hormones. (**B**) NIR BL intensities of the selected imaging probes in the lysates after stimulation using various steroid hormones. (**C**) Dose–response curves of LSSmNep-SH2-ER-NLuc (left) and LSSmNep-AR-NLuc (right) after stimulation of varying concentrations of steroid hormones. Inset ‘*a*’ illustrates the working mechanism of LSSmNep-SH2-ER-NLuc. Ligand-activated ER LBD is phosphorylated and recognized by the adjacent SH2 domain. The fold-up structure exerts enhanced NIR BL intensities. The p-values (student’s *t*-test) are ** < 0.01 and * < 0.1, respectively.

## Data Availability

The data presented in this study are available on request from the corresponding author.

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
