# Peer review of "Near-Infrared Imaging of Steroid Hormone Activities Using Bright BRET Templates"

_ijms, 2022, doi:10.3390/ijms24010677_

Round 1

Reviewer 1 Report

This manuscript entitled “Near-infrared imaging of steroid hormone activities using bright BRET templates” presents a development of novel variants of genetically encoded fusion proteins from luciferase and fluorescent protein for monitoring molecular events in cells based on near infrared bioluminescence resonance energy transfer (NIR-BRET). To select the optimal NIR-BRET pair, the authors studied in detail various combining pairs from five luciferases and 12 luciferins the most part of which are novel synthetic coelenterazine variants. Sometimes the study includes unnecessary details, rechecking the well-known information on the impossibility of cross-using specific bioluminescent substrates by coelenterazine-dependent and insect luciferases. The two best BRET combinations demonstrating efficient resonance energy transfer with a shift from blue to red were further utilized to develop new NIR-BRET systems for imaging steroid hormone activities inside cells.

The manuscript contains valuable information on comparing different bioluminescent reporters under the same conditions for intracellular expression, and on testing new substrate analogues of coelenterazine. Unfortunately, nothing more effective than the natural coelenterazine has been found so far.

The manuscript is well-structured and will undoubtedly be a useful tool for numerous researchers working in the fields of molecular and cell biology and using or planning to use bioluminescent reporters and bioluminescence imaging techniques in their work. Unfortunately, the work has yet some shortcomings.

Results and methods request more details.

 – As is known, disulfide-rich copepod luciferases of the type used in this work, mature effectively only in the endoplasmic reticulum. It is not clear from the manuscript what type of expression (cytoplasmic or ER) was used the BRET-templates with ALucs. This is important for interpreting the results.

 – Line 257. “…dissolved in universal buffers or assay buffers (Promega).” – There is no description of the composition or cited reference for “Universal buffer” and “assay buffer”. Since these are common names for many different buffers, it is then necessary to give the catalog numbers of the buffers used, and not just the company name.

– Line 364. “4.2. Measurement of BL spectra of CTZ derivatives according to marine luciferases.” – The cell density at transfection and the amount of added DNA are not indicated.

Minor remarks:

 – Line 16. “…NLuc-DBlueC and ALuc47-nCTZ combinations showed luminescence in the green emission…”. The emission with an optimum of 488 nm (ALuc47-nCTZ), especially with 458 nm (NLuc-DBlueC) cannot be named green, it is all blue light.

– Lines 23 and 239. Of course, LSS means “large Stokes shift”, not “large stock shift”.

– Line 127. “In this experiment, we intensively chose DBlueCs as substrates of NLucs instead of furimazines (FMZ).” – It should probably be like this: “…we intentionally chose DBlueCs…”.

 - Fig. 2 – What does a sign “D-luciferin” for the slide part with nCTZ from Fig. 2Ð’? Does this mean that both substrates were added? Also, there is no negative control in the experiment - cells without luciferase genes.

– Line 382. What does it mean “…absolute BL intensities…”?

Author Response

We have uploaded the response letter.

Reviewer 2 Report

 The authors of this study addressed the need of near-infrared shifted imaging tools through the development of bioluminescence resonance energy transfer systems. This was achieved by the analysis of various marine luciferases with coelenterazine analogues, that were subsequently paired to large-stokes shift red-light emitting fluorescent proteins. The application of bright and red-shifted BRET systems was furthermore shown to be a successful backbone for the study of steroid levels using intact and lysed mammalian cells.

Overall, the study is well devised and the article is well-written. The findings are of high interest to researchers studying molecular events in mammalian cells with implications to in vivo imaging. Specifically, the LSSmNep-Nluc system as a steroid biosensor backbone is highly interesting. The article should be considered for publication after addressing the following minor comments:

Comments:

1.       The study aim is a bit ambigous as the authors state (line 88) that the study intends to develop bright NIR-BRET templates without analysing BRET systems that incorporate Furimazine as the luciferase substrate. The authors later mention (line 126) that Furimazines were intentionally not used due to its bright luminescence hindering its use for multireporter systems. This might make sense if two BRET systems would be used simultaneously, e.g. in the same cell line, which appears to not be the case in this study. Please clarify.

2.       Please explain why NLuc, Aluc16 and Aluc47 were selected for this study (line 112).

3.       There are some grammatical mistakes in this manuscript.  Please correct. E.g.

line 127: ‘intentionally/deliberately’ not ‘intensively’

line 137: ‘…severely attenuated by haemoglobin and…’

line 177: ‘…shows a relatively weak BL…’

4.       The authors speculate that the detergents in Promega’s lysis buffer suppress the activity of NLuc. This seems highly unlikely that Promega is supplying its products with a basically non-functional lysis buffer. Can the authors back up this claim?

5.       The figures are generally hard to follow due to the abbreviations. Particularly, Fig.5  contains a number of abbreviations that are not explained in the legend. Please add to the figure legends to make it easier for readers to follow.

6.       The authors guess limits of detection for steroid sensing (line 339). Based on the presented data, it would be easy to use the linear part of the calibration data and calculate the detection limits (e.g. blank + three standard deviations of the blank). This would allow the comparison of steroid sensing with other detection tools.

Author Response

Uploaded the response letter.
